# Architecture of Medieval Armenia as a Field of Research for Russian and Italian Scholars: Comparative Analyses of the Historiography

**Armen Kazaryan**

Institute of Architecture and Urban Planning, Moscow State University of Civil Engineering (MGSU), 129337 Moscow, Russia; kazaryanayu@mgsu.ru

**Abstract:** For the first time in the literature, this study provides an analysis of the activities of two major architectural–archeological missions that investigated the architectural heritage of the Armenian Highlands: the Russian Ani Archaeological Expedition (1892–1893 and 1904–1917) and the Italian academic programs of the Universities of Rome and Venice and that of Milan Polytechnic (from 1966 to the 1980s). In this article, the results of the conducted research are compared, and their contribution to the development of the history of medieval architecture is evaluated. The differences in the results are related to the chronological distance between the missions, as well as the main focus of each work: the activities of the Russians are primarily archeological, while those of the Italian groups are architectural. The head of the Ani Institute, Nikolay Marr, set himself the task of exhibiting the original artifacts in the museum he had created in the medieval capital of Armenia, Ani, while the Italian professors relied on photography for both permanent and touring exhibitions. The second mission was in unspoken contact with the first, forming a kind of time-stretched dialog. Although, by the 1970s, almost none of the participants in Marr's expedition remained alive, his scientific works were periodically being published, with some still waiting their turn in the scientific archives.

**Keywords:** Armenian architecture; Ani; historiography; Russia; Italy; academic schools; geopolitics; cultural diplomacy





## 1. Introduction

This research aims to reveal an important feature regarding the history of studying the architecture of medieval Armenia, which presents a bright spot and a phenomenon in the history of world art. This feature lies in the activities of two separate research missions in the field of art history/architecture/archeology, with their virtual correspondence representing a cultural dialog. These two missions occurred in different historical periods and represented the Italian and Russian schools of medieval architecture, one of the most powerful schools of study in Europe.

In addition to stating the hitherto unnoticed fact of the existence of two major research missions focused on the territory of historical Armenia, the features of their activities are revealed. There is a plethora of information regarding each research mission, and, in recent years, many historiographical articles have been published, the purposes of which were to analyze not only the results of a Russian scientific project that ended more than a century ago but also the results of Italian projects that have already become the subject of recent history. Therefore, it is important to make a comparative study of the two initiatives to reach an understanding of the role of each of these missions in the subsequent development of this field of study regarding medieval art.

## 2. Projects, Research Groups, and Historical Facts

The architecture of Armenia, its monastic complexes, and, in particular, the ruins of Ani—one of the most brilliant cities of the Orient and the capital of the country from

961 to 1045 CE—attracted the attention of English, French, German, and Russian travelers of the 18th and 19th centuries (see Figure 1). From the 18th century onward, the Russian creative community was interested in Oriental antiquities within the framework of the pan-European trend and, from the middle of the 19th century, paid special attention to the monuments in Armenia and Georgia as the works of a branch of Byzantine architecture. Its development was considered important at that time from both the scientific and practical points of view since it contributed to the development of the academic Byzantine style in Russia (Baeva and Kazaryan 2021; Pechenkin 2021). Count Mikhail Semyonovich Vorontsov, the governor of the Caucasus in 1844–1854, drew scholarly attention to Ani and also sent a Russian army officer, Yu, to the site. In his six weeks of work at the settlement, Yu. Kestner drew 69 pictures of buildings and copied 42 inscriptions, which formed the basis of an album of 45 illustrated sheets (Leo 1963, pp. 39–40). Copies of the inscriptions were given by Vorontsov to the well-known Armenologist Marie-Félicité Brosset, who initiated a study of the monuments of Ani (Marr 1934, p. 10; Leo 1963, p. 40), a subject to which he eventually devoted a large monograph (Brosset 1860)[1].

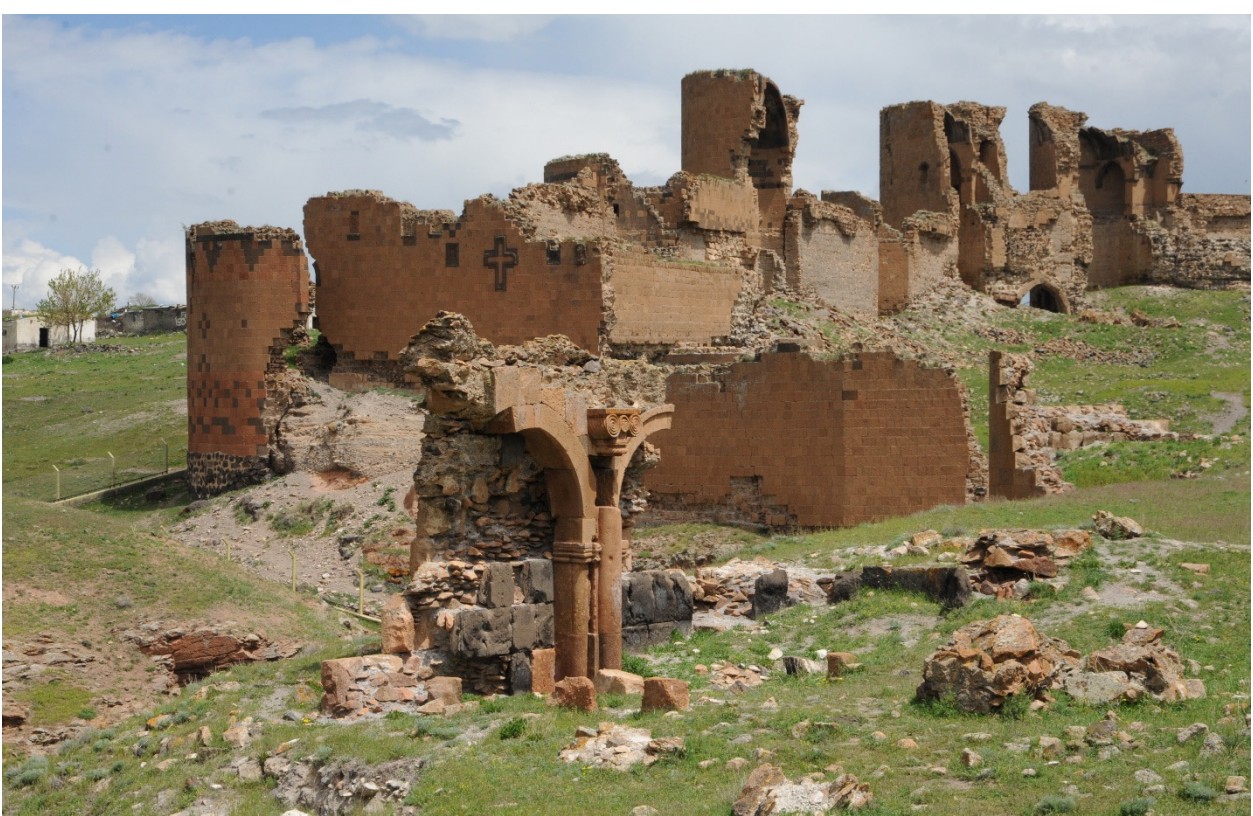

**Figure 1.** The western part of the city walls of Ani. Photo by the author (2023).

Ani became one of the most attractive centers of concentration of creative aspirations and pilgrimage, especially after 1878, when the territory of western Shirak Province, which had seen the settlement of Ani along with numerous churches and monastic complexes from the 5th to the 13th centuries, entered the borders of the Russian Empire and remained thus for 40 years. The apogee of the development of scientific research in Armenia was the work performed by the Ani archeological expedition, which, in 1910, was transformed into an institute (1892–1893 and 1904–1917) that was equipped by the Imperial Archaeological Commission. The expedition was led by the Orientalist historian and philologist Nikolay Yakovlevich Marr, a bright, highly educated person who rallied around him a team of Orientalists, art historians, architects, and students with various specializations. The main task of the expedition was to conduct archeological excavations (see Figure 2). However, the material itself—the urban structure, numerous ruined buildings, and the carved and

sculptural details—predetermined the focus of research as architectural, especially after the collective expanded its activities toward more practical issues regarding the preservation of monuments and the museumization of ruins and architectural fragments. The Ani Institute created a body of work comprising scientific research, excavations, measurements, the photographic fixation of monuments, museum curation, and publishing activities. Due to the fact that the research programs of the annual expeditions were not confined to the boundaries of the settlement, the Ani Institute became the only major academic center for the study of the material culture of Armenia and its architecture.

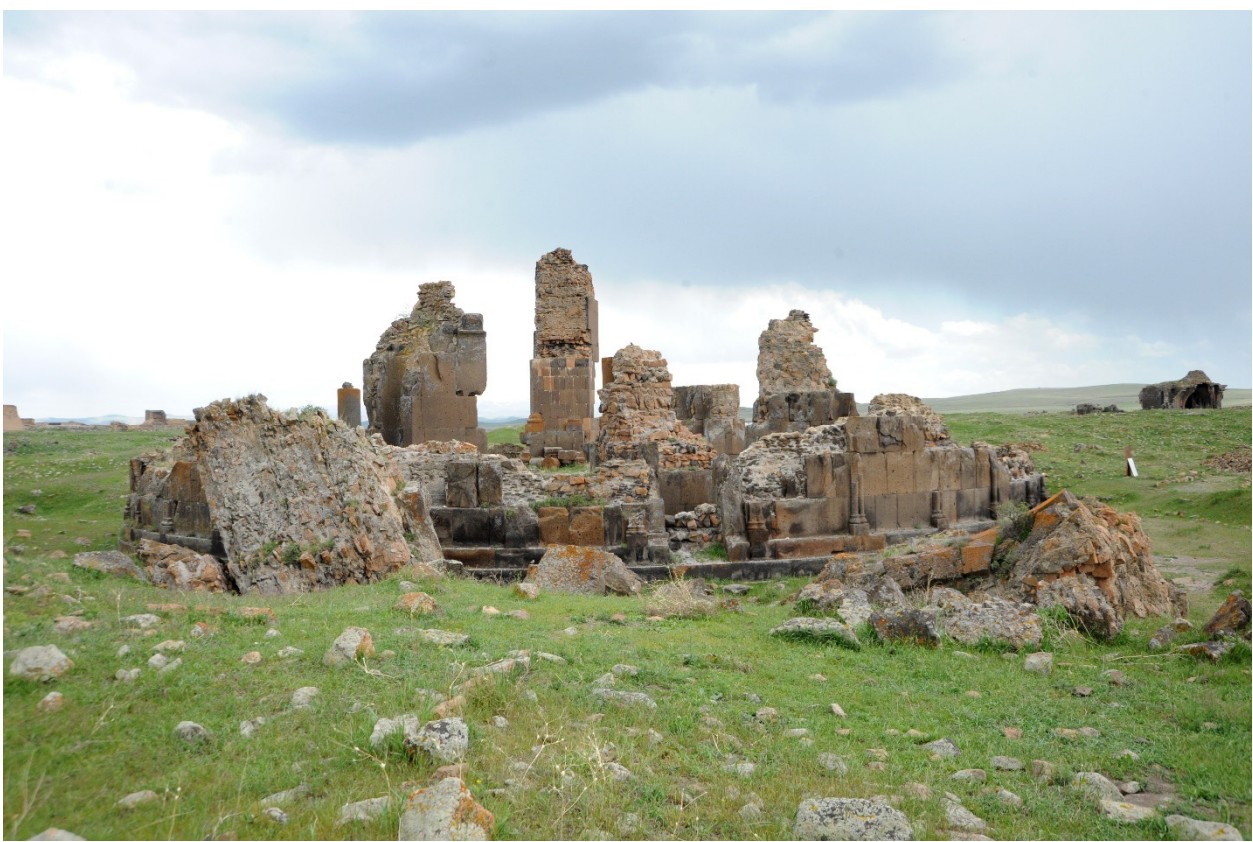

**Figure 2.** The church of Gagikashen in Ani, built in 1001 CE, which was excavated by Marr's expedition. Photo by the author (2023).

In 1903, the architect Toros Toramanian settled in Ani and began a systematic study of these monuments of Armenian architecture, capturing them in photographs and carrying out numerous qualitative measurements and graphic reconstructions, often in cooperation with the Marr expedition.

In the late 19th century, the settlement of Ani found itself within a Christian state, bringing the potential for consistent study. Russian archeologists had an opportunity to realize the site's potential after the Crimea, following the Caucasian direction of development and the introduction of the material values of Eastern civilization. The appointment of Marr as the organizer of this expedition was an absolute success, given his energy, charisma, and combination of the merits of a scientist with managerial qualities (Kazaryan 2016, pp. 12–13). Marr was a talented lecturer and had the gift of persuasion; the public funds supplied by the Armenian communities of Russia exceeded the amounts allocated by the Imperial Archaeological Commission several times over (Orbeli 1911, p. 14 (note); Kazaryan 2016, pp. 15–16). He founded 16 full-fledged archeological campaigns in Ani. The settlement quickly became not only a place of pilgrimage but also a tourist center: 2000 people visited it in the summer of 1917 (Marouti 2018, p. 123).

The closure of this institute in 1917 and the surrender of Ani to Turkey in 1918, followed by re-surrender in 1920, led to a prolonged cessation of research, the destruction of the Ani Museum created by Marr, and a threat to the preservation of architectural heritage, both on the site of this settlement and in its vicinity. The 1920 Armenian archeological expedition in the vicinity of Ani, organized by Ashkharbek Kalantar, witnessed the devastation of this region and reported no results that were comparable to the Marr expedition (Kalantar 2007, pp. 139–94; Ghazaryan (Kazaryan) 2015). The study of Armenian architecture continued in Yerevan, in other cities of the USSR, and in Western scientific centers, but the connection of historical and theoretical developments with archeological work and the full-scale study of monuments continued only within the territory of Soviet Armenia.

The structure supporting cultural heritage protection was also created there and limited research was carried out, notably the expedition of Nikolay Mikhailovich Tokarskiy to Amberd and the beginning of excavations in Dvin. The turning point came during the Second World War; one landmark event was the preparation for the publication of a collection of studies by Toros Toramanian. Then, in 1946, the first edition of Tokarskiy's monograph was published in Yerevan. A new generation of Armenian architectural historians came to the fore in the 1940s and 1950s, and, as a result, various studies were carried out over the next two decades.

The resumption of research within the territory of Eastern Turkey, while access was closed to Soviet scholars, was facilitated by various expeditions made by their Western colleagues from the very beginning of the 1960s. At the forefront of this new research were several Italian scientific groups that developed large-scale academic programs on the architecture of the Christian East. These communities of specialists were divided into two groups, led, on the one hand, by the Roman professors Geza de Frankovich, Fernanda de Maffei (Iacobini 2012), and Paolo Cuneo (Galdieri and Grabar 1997–1999), and, on the other hand, under the leadership of Professor Adriano Alpago Novello (Riccioni 2020a), who worked in Venice and Milan.

The first expedition was made with the support of Sapienza University in Rome and consisted of Tommaso Breccia Fratadocchi, Paolo Cuneo, Maurizio Guidi, and Gianluigi Nigro, taking place in the summer of 1966 in Soviet Armenia. Contacts were established with the Institute of Arts of the Academy of Sciences; dozens of monuments were inspected in many districts of the republic. In 1967, the expedition covered the northwestern regions of Iran and eastern Turkey, moving along the route south and east of Lake Van to the north, and on to the province of Kars. While based in Van, de Frankovich and de Maffei joined the group, which thus expanded on this trip. Over the course of two years, a considerable body of material was collected in all these historical territories of medieval Armenia (Bevilacqua and Gasbarri 2020, p. 30). The subsequent photographic exhibition on Armenian architectural monuments, which was mounted in 1968 at the Palazzo di Venezia in Rome and then moved to the Palazzo Ducale in Venice in July of the following year, left a strong impression on its visitors (Bevilacqua and Gasbarri 2020). A catalog was published that included an introductory article by de Frankovich, along with essays on Armenian art, medieval architects, and the features of Armenian architecture and the periods of its development, as well as annotations and plans of the area's 100 most important monuments (Breccia Fratadocchi et al. 1968).

The first expedition and the photographic exhibition stimulated further promotion and refinement of the program behind the multi-year project supported by the Conciglio Nazionale delle Ricerche. A number of valuable and fundamental studies conducted in subsequent years ended with publications forming part of the series *Studi di architettura medioevale armena/Studies on Medieval Armenian Architecture* (Conciglio Nazionale delle Ricerche. Director: Professor Geza de Frankovich from the University of Rome). In parallel, work was carried out on a detailed documentation of Armenian architecture. The brilliant result of these endeavors was the catalog of all monuments of medieval Armenian architecture that were known in the 1980s within the boundary of the Armenian Highlands (Cuneo and Breccia Fratadocchi 1988).

No less fruitful was the activity of the group headed by Adriano Alpago Novello. In the series of books titled *Ricerca sull'architettura armena* (Milano 1970–2005), the research programs were published, providing the full list of monuments, a bibliography, the results of several research studies, and information regarding the state of buildings and restoration activities. The last book in the series was the catalog of the 2004 exhibition (Alpago Novello 2005). *Documenti di architettura armena/Documents of Architecture of Armenia* (23 vols, 1968–1998, Milano: Edizioni Ares; Venezia: OEMME Edizioni) comprised a series of published monographs on the most outstanding works of Armenian architecture, with high-quality measurements and photographs. During this period, expeditions and exhibitions were actively being conducted. The first exhibition was held in October 1968 at the Brera Pinacoteca in Milan; over the next 15 years, this exhibition visited 18 countries, eventually moving to permanent storage at the Yerevan Museum of Architecture (Riccioni 2020a, p. 214; Riccioni 2020b, pp. 18–19; Spampinato 2020). Firstly, the exhibitions served as a tool for popularizing the scientific project, and secondly, they offered an opportunity to present the results of trips to a wide-ranging environment and promote them at an international level.

Thus, from the 1960s to 1980s, the second period of systematic collection of scientific material began in eastern Turkey and, at the same time, in Armenia as a result of the cooperation of Italian scholars with their colleagues from Soviet Armenia. The architect Armen Zarian, who moved from Rome to Yerevan in 1963 (Ieni and Zekiyan 1978), made an outstanding contribution to the development of such cooperation with the organization of large-scale international symposiums on Armenian art, which have been held regularly since 1975 in the cities of Italy and Yerevan (Zarian 2009).

In those years, Italian scientists collaborated with historians of medieval architecture from Moscow and St. Petersburg, especially with Hovhannes Khalpakhchian, whose interests included research into Ani and Akhtamar based on archival materials from Marr's Ani expedition. Paolo Cuneo and other Italian colleagues repeatedly visited the All-Union Research Institute for the Theory of Architecture and Urban Planning (VNIITAG, now NIITIAG), where Khalpakhchian headed one of the scientific departments. He had a great appreciation for the books and collections of articles published by the Institute and hosted its staff at some conferences in Rome[2]. It should be noted that Fernanda de Maffei maintained close contact with colleagues from the State Institute of Art Studies until the 2000s.

Scholars from other European countries have also shown interest in Armenian architecture. Of particular interest are field studies of the monuments published by Michel and Nicole Thierry, who began conducting expeditions to various regions of western Armenia in 1964. Cycles of published articles, as well as monographs by Michel Thierry on the monuments in the Kars, Van, and other vilayets of Turkey contain extensive material and sometimes serve as the only record of the condition of monuments in eastern Turkey in the period up to the 1990s. However, in terms of research objectives and the number of published monographs, the works published by the Italian groups of the 1960s and 1970s had no equal (Marouti 2018, pp. 175–83).

In fact, it was the Italian scientists who were the true successors of the tradition laid down by Marr. What is the reason for such a conclusion? What is the similarity between these two periods, 40 years apart, each of which lasted a little more than a quarter of a century? I will try to answer these and other related questions in this article.

Simultaneously with de Frankovich's retirement in 1972 and the dissolution of the research groups in Rome and Milan, there was a general weakening of interest regarding Armenia and, above all, the termination of research expeditions in these areas due to increasingly difficult political conditions (Bevilacqua and Gasbarri 2020, p. 42). In the successful expeditions of the 1960s and 1970s, Mariuti sees the pivotal role of individuals such as Armen Zaryan and Alpago Novello, along with the activity of their Italian colleagues, and also suspects that this activity reflects "the closeness between the Italian socialist parties and the Soviet regime in Moscow" (Marouti 2018, p. 176). While fully

agreeing with the first assumption, it is possible to suggest a relationship between this academic process and the political circumstances, but this finding is in tandem with the formulation of other issues that are outside the framework of this research. Nevertheless, the formation of the independent Republic of Armenia in the early 1990s contributed to the implementation of closer direct agreements with Italian organizations in the fields of monument protection and education.

## 3. Brief Historiographical Overview

Much has been written about the Marr expedition of the late 19th and early 20th centuries, as well as the campaigns of the Italians in the 1960s–1980s. These are short works, articles of an informational and analytical nature. The various assessments of these two expeditions are very different, in terms of their importance to the development of research on Armenian architecture, and, in some studies, giving priority to any one of them. In a brief historiographical essay on the achievements and prospects for research in this field, Murad Asratyan highlights the roles of two scientists, Joseph Strzygowski and Toros Toramanian, during the first stages of studying Armenian architecture. The results regarding the work of the Marr expedition, which began a decade before Toramanian's arrival in Ani and a quarter of a century before Strzygowski's visit to Armenia, were not actually evaluated as being significant (Hasratyan 2003, p. 474). It is as though two of our European colleagues neglect the era of Marr's Ani expedition when they conduct their own historiographical analyses (Foletti and Riccioni 2018).

Meanwhile, based on the achievements of these three indisputably outstanding scientists—Marr, Toramanian, and Strzygowski—by the middle of the 20th century, basic theories regarding the history of Armenian architecture and monumental art were being developed in Armenia, Russia, and the West. The seminal works of Y. Baltrusaitis, N.M. Tokarskiy, V.M. Harutyunyan, O.H. Halpakhchian, and S.H. Mnatsakanyan were based on the materials and ideas of these three pillars of composition of the research tradition. The theoretical basis and skills of Marr and Strzhigovsky were based on the traditions of the St. Petersburg Archaeological School and the Vienna School of Art Criticism, respectively. Toramanian, however, was an architect and, on the one hand, provided invaluable assistance in the work of the Marr expedition, while on the other hand, he provided graphic and photographic materials to Strzygowski, which were published in the latter's own book. The role of Marr and, possibly, other members of the expedition in the nurturing of Toramanian as an architectural historian seems to be undoubted, as can be concluded from correspondence written at the time. Strzygowski also communicated not only with Marr but especially with a member of the Ani expedition, the famous Byzantinist Yakov I. Smirnov. Both of them shared their photographs with the Austrian professor (Khrushkova 2013, pp. 570–74) and, undoubtedly, their thoughts.

All this makes it possible to identify the status of these three figures, who were primarily associated with the beginnings of the study of Armenian architecture, while taking into account a mutual exchange of ideas, even if they contradicted each other, as well as with the help of material in their works. The Ani expedition was the focus of attention of Russian Oriental studies and offered common ground for international scientific communication.

For many years, especially for Europeans, studies on Marr's Ani Expedition remained in the shadows, unnoticed against the background of Toramanian's essays, measurements, and reconstructions (Toramanian 1942–1948; 2008). This was mainly due to the monumental nature of Strzygowski's book, "Architecture of Armenians and Europe" (Strzygowski 1918). Another reason was the abrupt interruption of Marr's research as a result of the closure of his Institute and the capture of Ani by the Turkish army. Neither Marr nor his colleagues continued these studies for a considerable time; many of their results remained in manuscript form and only began to be published after the Second World War, while some are still waiting for their turn. Marr only summarized the results of the expedition in

his 1934 monograph of the city of Ani, which did not include the works conducted in the last three years, and which mostly contained archeological material (Marr 1934).

For Europeans, the only scholar who was known to explore Armenian architecture for decades was Strzygowski. Today, when it comes to the roots of the tradition of studying Armenian architecture founded in Italy in the 1960s, this phenomenon is certainly associated with the controversy that came to pass in this country in connection with the provocative conclusions of an Austrian professor about the genetic roots of the compositions of European cathedrals (Bevilacqua and Gasbarri 2020, pp. 26–28)[3]. Strzygowski's theoretical views formed the basis of most studies by Armenian scholars (Bock 1983, pp. 112–18; Kazaryan 2012, vol. 1, p. 43), who, nevertheless, consider Toramanian the patriarch of the history of Armenian architecture. In his research, they found the answers to many questions regarding periodization, the stylistic development of architecture, and the reconstruction of monuments (see Figure 3). At the same time, when considering the development of architecture in a broader context and taking into account the historical and cultural components of such development, in some studies, for example, in Tokarskiy's books, a reliance on the heritage of Marr is clearly felt (Tokarskiy 1961). We share the approach of incorporating joint consideration of the activities of these three personalities, as manifested in a dissertation by Andreh Marouti, who commented: "The first comprehensive scientific study of Armenian architecture was published in 1918 by Josef Strzygowski, based on the materials provided by Toros Toramanian and the systematic excavations by Nicholas Marr at Ani between 1904 and 1917" (Marouti 2018, p. 1 of the Abstract; see also p. 144).

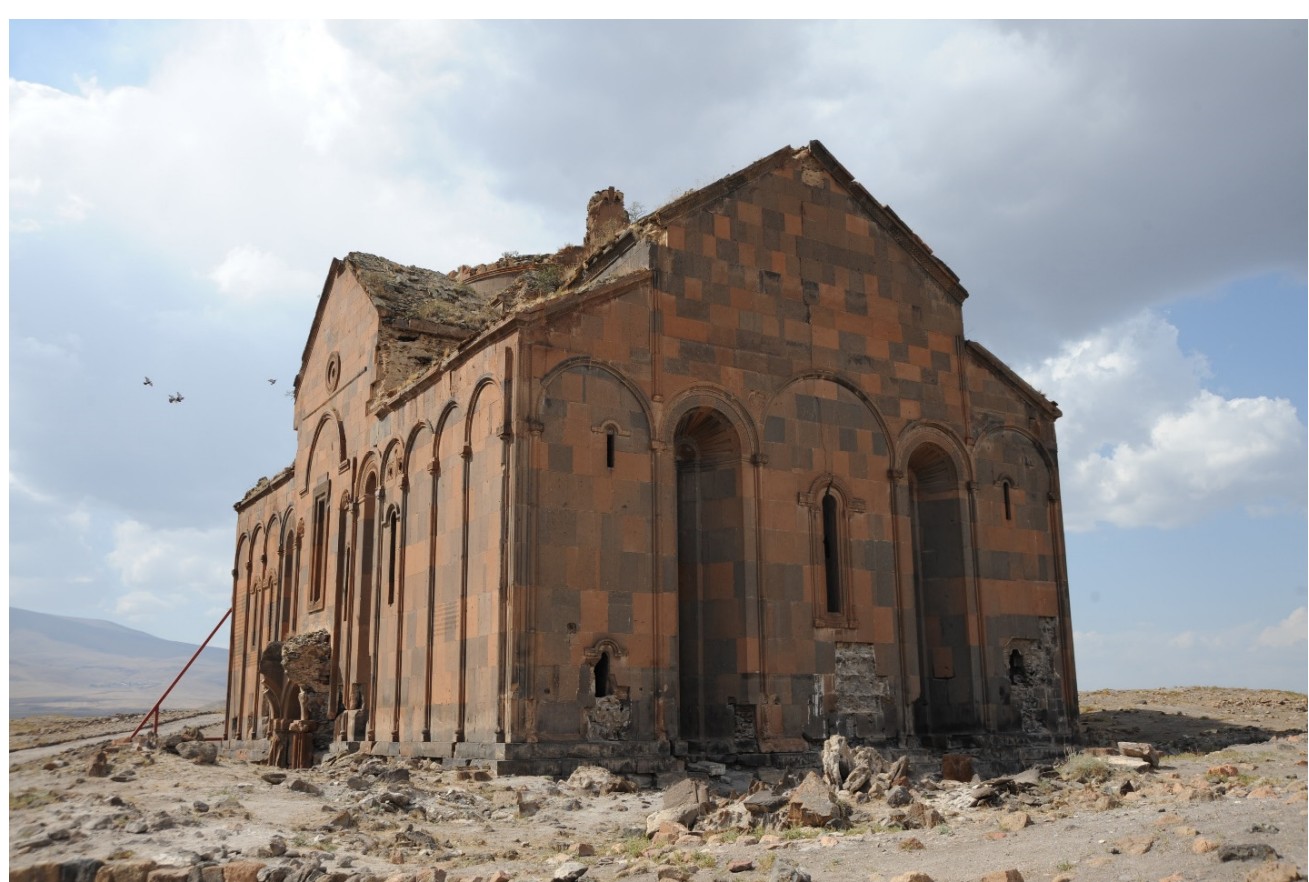

**Figure 3.** The cathedral in Ani, seen from the southeast. Photo by the author (2012).

V.A. Mikhankova made a significant contribution to the identification of the materials of the Ani Expedition and analyzed the activities of the Marr expedition (Mikhankova 1949). Back in the 1960s, Stepan Mnatsakanyan prepared and published a monograph titled "Nikoghaos Marr and the Armenian Architecture" (Mnatsakanyan 1969), which clearly

articulates the main achievements of the great archeologist in the field of architecture studies. Focusing mainly on the architectural monuments of Ani, Marr, following the handwritten sources, gave them primary importance for understanding the development of the area's history, culture, and social relations. The results of his study refuted a widespread opinion regarding the onset of the decline of Armenian architecture in connection with the fall of the Bagratid kingdom, revealing to the scientific world the brilliant flourishing of architecture in the Zakarid era, which came at the end of the 12th century. According to Mnatsakanyan, Marr was the first to attempt to consider Armenian architecture in the context of the art of a vast region and, by noting the diverse influences of neighboring traditions, he drew attention to the need to identify the local roots of architecture, especially regarding the constructive essence of buildings (Mnatsakanyan 1969, pp. 172–80).

Some remarks regarding the activities of Marr, Orbeli, and other participants of the Ani expedition can be found in the latest research on the work of the Imperial Archeological Commission and the archeologists of the St. Petersburg school. The work of the Ani Expedition was also discussed in articles on Marr's associates. For example, in the book by Karen Yuzbashyan and in the article by Babken Arakelyan commemorating the 100th anniversary of the birth of Iosif Abgarovich Orbeli, the authors discuss the participation of this scientist in the Marr expedition, his role as a director of the Ani Museum of Antiquities (since 1908), as the author of *A Short Guide to the Settlement of Ani*, published in 1910, and *The Ruins of Ani* from 1911, his leadership of archeological work in Garni in 1909–1910, and his subsequent business trips to Khachen (Nagorno-Karabakh) and Western Armenia, which ended with the publication of important articles (Yuzbashyan 1987; Arakelyan 1987). A profound analysis is made in the article by A.Ya. Kakovkin of the research of Nikolai Petrovich Sychev, who, in 1911–1912, examined three Ani constructions: Bakhtagek's Church, the Palace Church, and the Church of St. Gregory or Tigran Honents. His conclusions about the nature and dating of the murals in the church of Honents and its *gavit* (narthex) have not yet lost their relevance (Kakovkin 1987) (see Figure 4). Nikolay Lvovich Okunev's research at Ani is discussed in Yulia Yancharkova's monograph on the scholar's legacy (Yancharkova 2012).

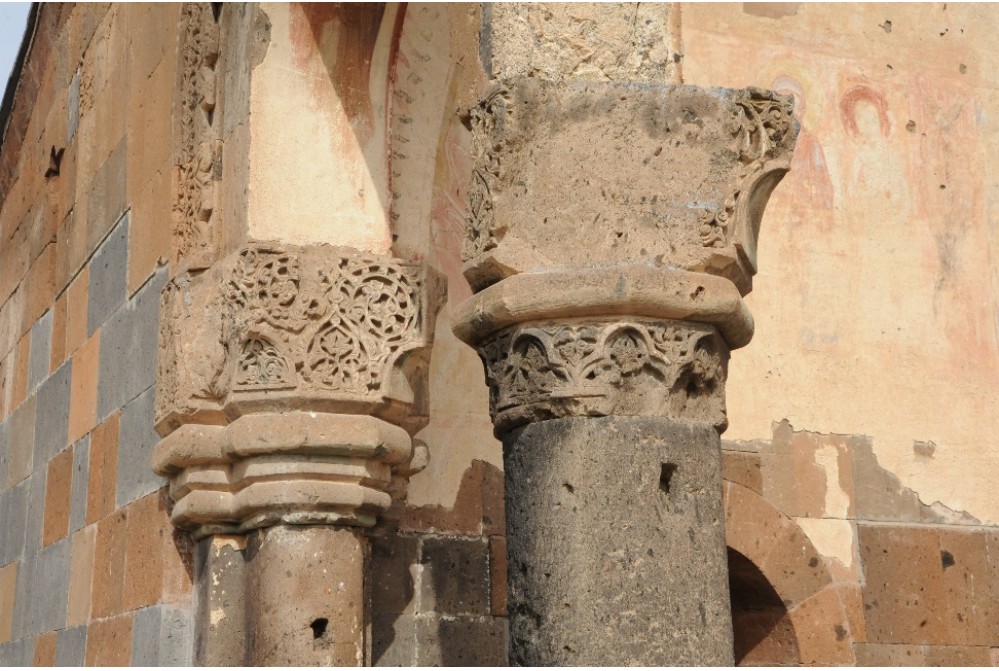

**Figure 4.** The zhamatun or gavit of the church of Tigran Honents in Ani, showing the capitals. Photo by the author (2012).

According to the Ani expedition program, research into the so-called underground Ani, with its numerous rock-cut residential and utility rooms and chapels, was carried out by a younger participant named David Kipshidze, whose materials had already been published by Tokarskiy in the 1970s (Kipshidze 1972). The results of these works were evaluated in recent years by a group of Italian speleologists and their Turkish and Russian colleagues (Bixio et al. 2009; Yazıcı 2017; Baeva and Klyuev 2022; Baeva and Kazaryan 2022a) (see Figure 5).

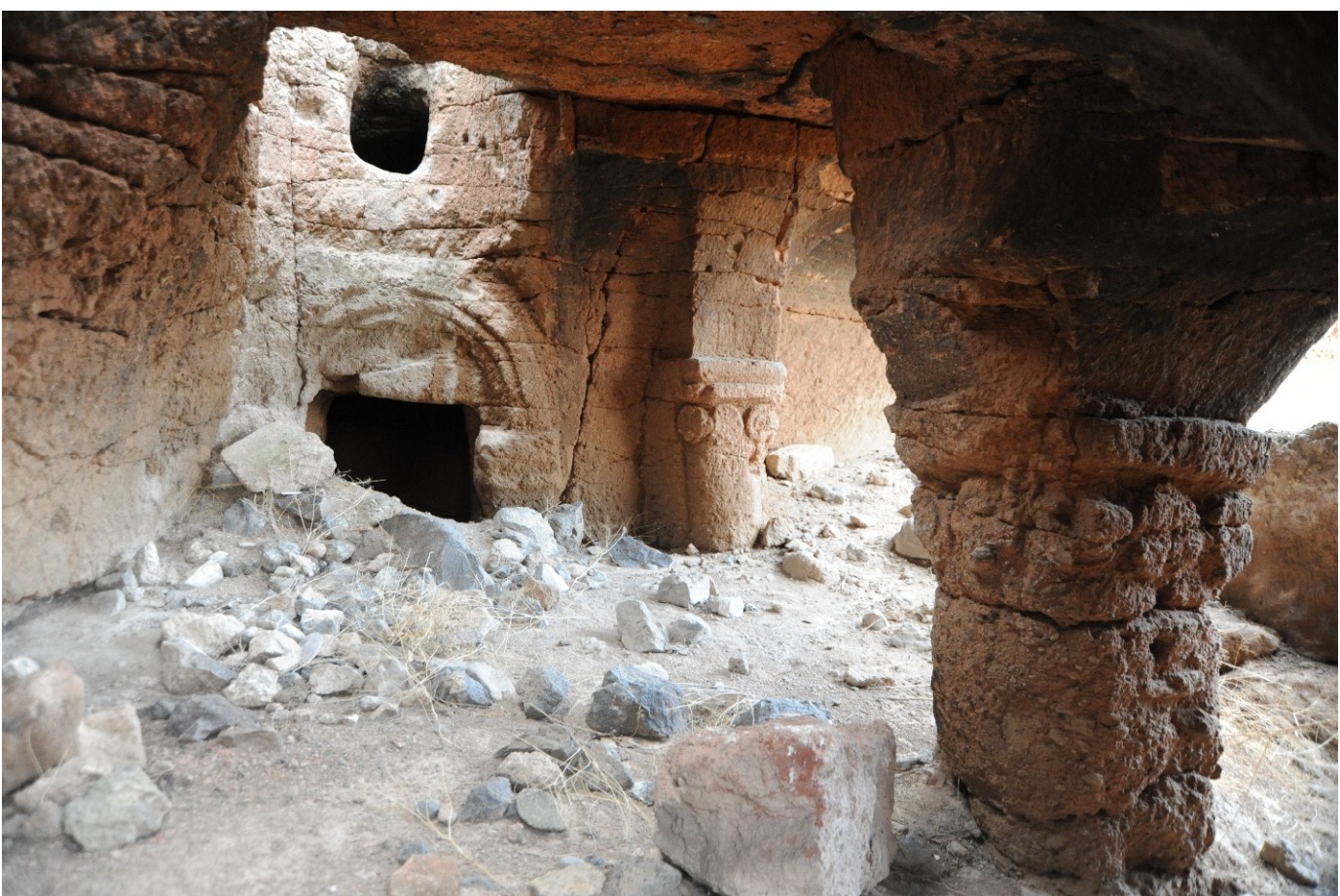

**Figure 5.** The rock-cut monastic complex in the Gailadzor gorge in Ani. Photo by the author (2022).

The topic of Armenian archeology is steadily becoming more and more popular in different countries, as evidenced by the latest research published by our colleagues from Armenia, Italy, France, the USA, Turkey, and Georgia. Due to increased interest in the results of the Marr expedition, Russian scientists have now published a series of historiographical studies. Their goal is not only to identify the features of the Ani expedition but also to determine further ways to study the architecture of this city, in connection with the development of such research in Russian scientific centers (Baeva and Klyuev 2022; Baeva and Kazaryan 2022b; Kazaryan 2022; Kazaryan et al. 2022).

To popularize the subject matter, as well as to attract attention to the work of the Marr expedition, exhibitions were organized that were composed, either entirely or partially, of photographs from the late 19th to the early 20th centuries, titled "Ani: The Millenial Capital of Armenia", displayed in the History Museum of Armenia (Yerevan 2015) (Grigoryan 2015) and in a series of exhibitions titled "The Poetry of Stones: Ani, An Architectural Treasure on a Cultural Crossroads" (Istanbul 2018; Ankara 2018; Yerevan 2018; Oslo 2019)[4].

The original Italian expedition to Armenia was accompanied by numerous news reports and, for the last quarter of a century, it has been analyzed in both historiographical

and popular scientific articles[5]. Undoubtedly, there was a need for historical distance and the arrival of a new or partially new generation of researchers, along with the emergence of new interest in the problems of medieval Armenian architecture. The extinction of this interest at the end of the 20th century and its almost complete absence in the first decade of the 21st century provided that distance, after which, on Italian soil, there was a need to revive, if not the tradition, at least the idea of an "Armenian" project. During this pause in interest, the initiative moved to other countries where, in response to the need for fundamental research and a theoretical understanding of the material that had been collected for decades, individual initiatives and projects were implemented—by Anegret Plontke-Luning in Germany, Patrick Donabédian in France, Christina Maranchi in the USA, and the author of this article in Russia (Plontke-Lüning 2007; Donabédian 2008; Kazaryan 2012; Maranci 2015). Art historians and archeologists from these countries are still working on major studies on Armenian architecture, in cooperation with their colleagues and scientific centers based in Armenia and Turkey.

Disclosure of the potential of Italian studies takes place within the framework of the revival of research at the University of Ca' Foscari, whence Alpago Novello transferred the Center for the Study and Documentation of Armenian Culture in 1991. Interest in the history of the study of Armenian architecture, especially in the activities of the missions of Italian scientific groups, seems completely justified. All the significant historiographical articles in Venetian publications have been published in the last 5 years. Especially valuable was the collection of reports titled *Studies in Armenian and Eastern Christian Art*, edited by Aldo Ferrari, Stefano Riccioni, Marco Ruffilli, and Beatrice Spampinato on the results of the conference in Venice (Ferrari et al. 2020). Most of the articles are devoted to the research work of the Italian scientific tradition of the 1960s and 1970s and contain a selection of statistics, along with a critical analysis of the development of research. The widest possible geography of research is touched upon in Marouti's dissertation (Marouti 2018), which was defended at the Politecnico di Milano in 2018.

Scholars are particularly interested in why research began in the 1960s. To this end, we examine the interwar years and the domination of fascist ideology. Researchers agree that this interest in Armenian architecture found in Italian academic circles was undoubtedly fueled by Strzigowski's inherently provocative concept of the role of the East and, especially, of Armenian architecture as a source of ideas for European medieval styles and central dome structures. In their attempts to defend the Roman roots of European architecture, Italian scholars needed to better acquaint themselves with the peculiarities of Armenian churches and determine the origins of their architectural forms themselves. From recent historiographical studies, we learn that Professor Pietro Toesca, who rejected the propaganda of the regime, who paid attention to the art of the East, and who collaborated with Geza de Frankovich in writing an encyclopedic article about Armenian architecture, turned out to be the most suitable leader for the emerging Roman group of researchers into Armenian architecture, many years later (Riccioni 2020b, p. 17; Bevilacqua and Gasbarri 2020, pp. 26–29).

In a number of historiographical articles, the history and principles behind the organization of exhibitions by the Roman and Milanese groups have been thoroughly investigated, while in others, the author's attention is focused on the nature of scientific research and the preferences of individual scientists.

It is noteworthy that it was mostly Russian and Armenian historiographers who wrote about Marr's expedition, while scholars from Italy and Armenia wrote about the essence and significance of Italian studies of Armenian architecture. Researchers from other countries touched on these topics casually and without having a direct scientific interest in them. Of course, there are some exceptions to this rule. It seemed interesting to me to compare the results of scholarly activity from these two periods. For me, this is the only way to understand the features and advantages of each of these two outstanding scientific and humanitarian missions, which were projects with an international scope.

## 4. The Scope and Results of the Work of the Ani Archeological Institute

It is necessary to emphasize the complex nature of the Ani Expedition/Ani Institute. Firstly, the variety of research that was carried out affected all spheres of the development of spiritual and material culture, as well as all aspects of architectural creativity: residential quarters, small houses and palace buildings, hotels and baths, industrial complexes, churches and monasteries, fortifications and engineering structures, and complexes formed of cave structures. Secondly, the expedition was distinguished by the breadth of its activities. Archeological excavations and the fixation of monuments were its main focus. Taking into account the significant volume of finds of the first companies of 1892 and 1893 and the number of valuable artifacts scattered around the settlement, Marr perplexed the Imperial Archaeological Commission with the idea of creating a museum as a place of storage and the scientific processing of architectural fragments, samples of sculpture, carvings, epigraphy, and household items. It was only after achieving this goal in 1904 that he resumed his work on the expedition. The task of preserving the architectural monuments in the settlement of Ani became a priority in the 1910s. Experience of the first restoration practices conducted on the buildings of medieval Armenian architecture was acquired by the Marr team in Ani on a number of the most famous monuments, such as the Churches of the Redeemer and of the Abughamrents family, along with the Gavit of the Church of the Apostles[6] (see Figure 6). Collecting information and cataloging monuments, including architectural monuments, as well as publishing activities to pursue both scientific goals and the popularization of architectural heritage, were a matter of special attention for the leader and the main participants of the expedition.

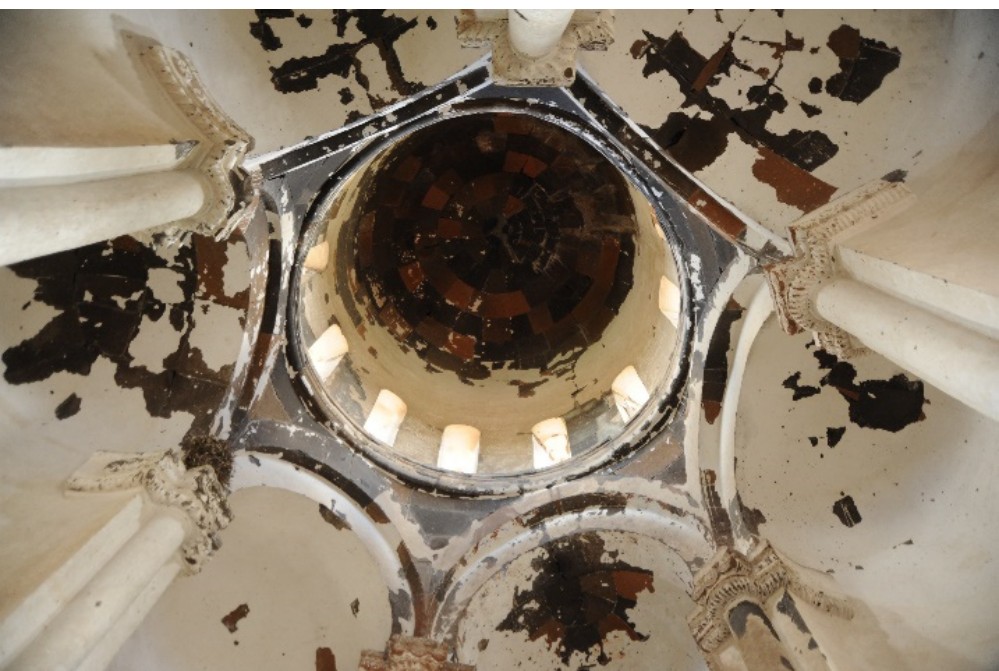

**Figure 6.** The Abughamrents family church in Ani; interior. Photo by the author (2017).

The work of the expedition, which was based in Ani as the main center for the development of medieval Armenian culture, was never confined to merely the monuments of this particular settlement. The work conducted in the first two years (1892–1893) covered a significant part of the Erivan province and the Kars region of the Russian Empire. Later, a study of the architecture of Shirak was accompanied by periodic trips to villages and monasteries in the vicinity of Ani, and an interest in ancient periods of history attracted their attention to the heritage of the Ararat Valley, to the organization of excavations of an ancient temple in the fortress of Garni, and to the study of the monuments of Artsakh, Vaspurakan, Taron, Taik, and Klarjeti. Marr showed particular interest in the prehistoric Vishap steles

erected in locations with water sources. All this research expanded the possibilities of drawing up a general picture of the development of Armenian culture and contributed to solving various issues regarding periodization, the geography of the development of Armenian architecture, and the identification of monuments. Marr entrusted most of the work conducted outside the settlement of Ani to other expedition members, namely, I.A. Orbeli, Ya.I. Smirnov, N.L. Okunev, N.M. Tokarskiy, and others. Realizing the importance of the concentration of the most outstanding buildings, in an architectural sense, that could be found in Ani, N.Ya. Marr sought to explore the heritage of the ancient city in the broader context of architectural monuments and urban culture. Achieving the maximum possible geographical latitude and chronological depth of research is a quality that was consistently attained in subsequent expedition seasons.

Marr's desire to consider works of art and architecture against a broader background of Middle Eastern cultural traditions, while taking into account global historical changes in specific epochs, contributed to building an objective picture while taking into account the social environment of the customer of the palaces, churches and fortifications and the existence of material values.

Marouti rightly calls the transformation of the organizational side of Russian archeology and the appointment of Marr as the first director of the Russian Academy of the History of Material Culture, formed after the revolution, "the shift from antiquarianism to attention to material culture", and he also mentions its result: "The academy funded studies and preservation projects regarding vernacular, civil, non-religious, non-monumental, and mundane aspects of human life. The result is publishing various monographs in further years and scientific preservations of non-monumental structures all around the USSR" (Marouti 2018, p. 117).

We cannot doubt that Marr's expedition expressed the sentiments felt by the Russian intelligentsia, and its equipping by the Imperial Archaeological Commission could also pursue the geopolitical goal of expanding their archeological and scientific knowledge of the lands annexed by the gradual expansion of Russia's borders throughout the 19th century, incorporating the cultural heritage of antiquity and Byzantine art in its orbit. It is noteworthy that the medieval heritage of the Georgians and Armenians was represented first by the European, and then by the Russian academic school, as a part of Byzantine architecture. Marr's intellect and the multinational composition of his expedition staff provided a critical rethinking of this assumption, leading to conclusions about the merits of Armenian and Georgian art, especially in the context of church architecture (see Figure 7). In this regard, it is impossible to treat some of the current ideas regarding pre-revolutionary Russian science, in line with the denial of the national roots of art and the evaluation of the artistic creativity of the Caucasus, as a peripheral phenomenon of the Middle Ages— peripheral, that is, within the framework of Byzantine art (Foletti and Riccioni 2018, p. 10). Not only in response to Toramanian's views but also, in many ways, to Marr's ideas, researchers investigating Armenian and Georgian architecture won the right to present these phenomena as completely independent entities.

Ekaterina Pravilova notes the coincidence of the timing of the Ani expedition with an increased scientific interest in urban culture and the emergence of a cultural mythology of cities, which informed the particular vision of Ani's history developed by Marr. Referring to Marr's essay, published in a famous collection of works by the Russian intelligentsia in support of Armenians (Marr 1898), she notes that "... Marr tried to gently reroute the narrative of Ani's rise and decay by extending the range of themes and evidence and highlighting the periods of growth in the times when Ani fell under the authority of foreign rulers. The paradoxical upshot of Marr's research was that Ani continued to flourish as an Armenian city even during the periods of political dependence. This was one of Marr's most important principles: political sovereignty has no direct influence on artistic production or cultural and economic development" (Pravilova 2016, p. 83). Now, looking back at the decades of development experienced by Soviet Armenia, including the development of its national culture, we can probably agree with Marr's assessment of some periods of the

country's development in the Middle Ages. Some contradictions in Marr's statements have been revealed in the article by Pravilova; their evolution toward cosmopolitan tendencies seems to me to be inevitable, with such a confluence of historical circumstances and the ongoing political situation in pre-revolutionary Russia. However, in general, Marr retained the dignity of an objective scientist and showed courage in highlighting certain problems, some of which remain poorly studied and are extremely relevant in our own time. Andreh Marouti's assessment of the significant role of this scientist is indisputable: "Being free from nationalistic prejudices, he did a more comprehensive study and educated the next generation of architects from Russia, Armenia, and Georgia" (Marouti 2018, p. 114).

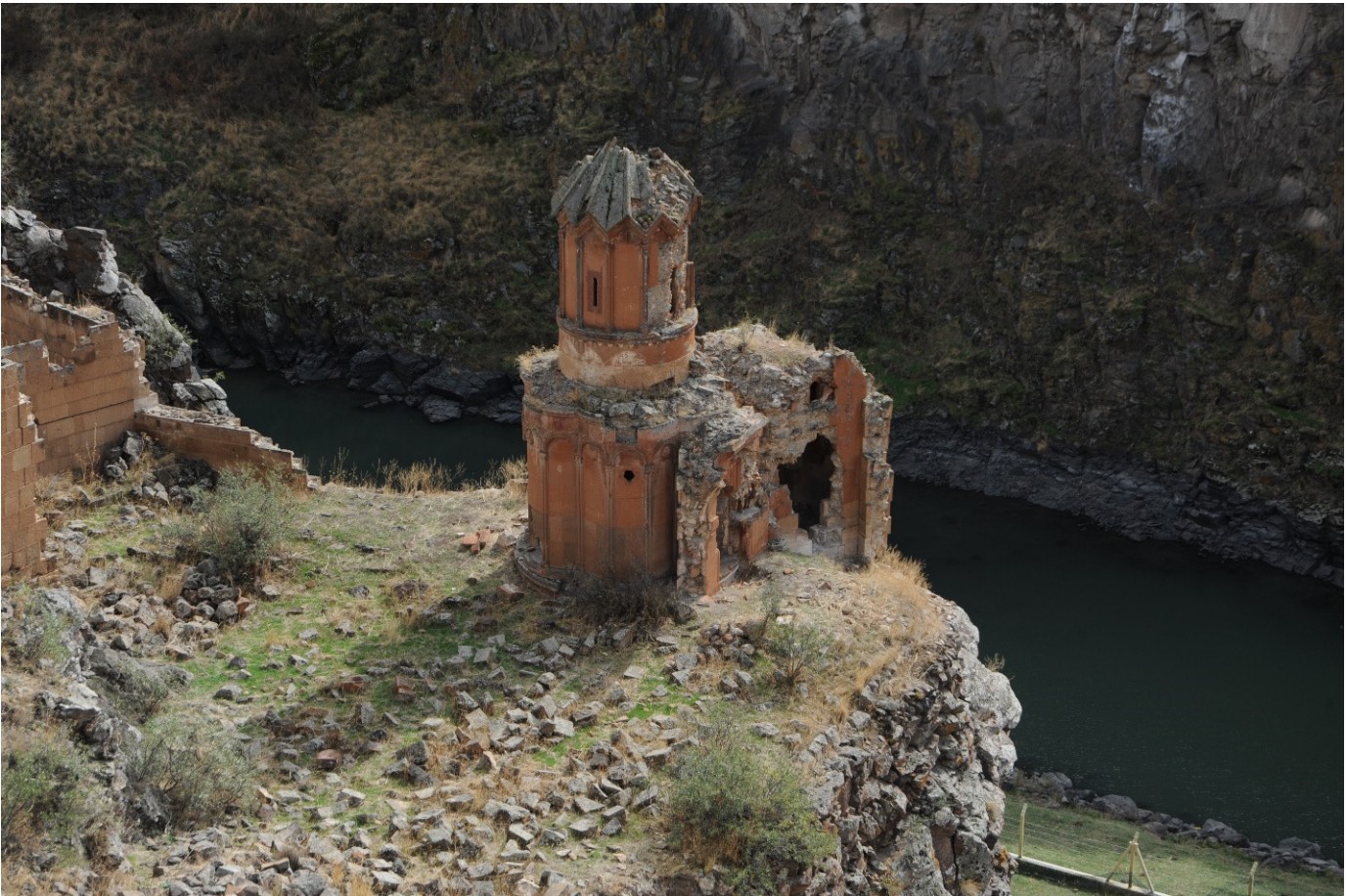

**Figure 7.** Kusanats vank in Ani. Photo by the author (2022).

Similarly, based on the study of a large number of architectural monuments and written sources from the 12th–14th centuries, the task was set to determine the position of the Ani monuments in relation to the architecture and monumental art of the Muslim world and, primarily, of Iran. Later, historical events led to the need for caution by academics exploring this matter and the actual cessation of research into this topic for many decades.

Questions were raised at the very beginning of Marr's tenure about the relationship between Armenian and Georgian architecture—debate on this issue was still to be resolved in the second half of the 20th century, between scientists who were present at the origins of architectural science in Armenia and Georgia, then two republics of the USSR. Marr's comprehensive study of the Chalcedonian Armenian culture, which ensured close contacts with the Byzantine and Georgian traditions, had already laid the foundations for a competent assessment of the historical role of this ethno-confessional group in the development of medieval architecture throughout the entire Caucasus region. It is on this basis that it will shortly be necessary to reveal the special character of several monuments in Ani built in the first half of the 13th century.

**5. The Scope and Results of Italian Research Programs on Armenian Architecture**

The work of the Italian groups is also characterized by a complex, multi-vector approach. Unlike the Russian team, the researchers did not set up a permanent location in the Armenian Highlands but instead enjoyed the hospitality of the state scientific institutions of Yerevan, mainly the Institute of Arts of the Armenian Academy of Sciences.

The expedition programs were carefully planned, and it was for this purpose that the lists of monuments were compiled. It should be borne in mind that most of the monuments of Armenian architecture in Turkey are not registered even now by the cultural heritage protection authorities. At the time of the original programs, the accounting system in Armenia was just being adjusted, and later divisions for the architectural monuments were created. Books contained the majority of information—at that time, the most recent publications were works by Edouard Utudjian (1962) and Nikolai Mikhailovich Tokarskiy (1961) (Riccioni 2020a, p. 206). It was necessary to first understand the volume of the most significant buildings, classify them according to different characteristics, and form an initial idea of their dates.

Trips were organized around the collection of information, in the form of photographs and some measurements of the monuments. Technical equipment, photographic equipment, and high-quality film recordings of the expedition were at the highest European level, and, for the first time, every building and architectural fragment was photographed in all possible detail. Painstaking lists of the literature were also compiled. This approach of collecting and cataloging material for the program of the Roman group ended with a large project and the publication of the first-ever catalog of Armenian architecture, published in 1988 in two volumes and edited by Paolo Cuneo, in collaboration with his Roman and Yerevan colleagues, namely, Tommaso Breccia Fratadocchi, Murad Hasratyan, Maria Adelaide Lala Comneno, and Armen Zarian (Cuneo and Breccia Fratadocchi 1988). It is noteworthy that the Milan group, which dealt not only with Armenian but also Georgian architecture, published a little earlier, in 1980 in Brussels, producing a large catalog on the art and architecture of Georgia, edited by Alpago Novello with the participation of his European and Georgian colleagues.

The tasks of cataloging, which were the main focus of the activities of both Italian groups, are part of the mainstream of the scientific activities still performed by Medievalists and Byzantinists in the last third of the 20th century and at the beginning of the 21st. There are two reasons for this emphasis. Firstly, the number of generalizing and analytical studies that put forward ideas regarding the development of medieval architecture has greatly increased in the previous few decades, and the reliability of most of the concepts at the time seemed doubtful and required confirmation with respect to a large number of monuments. This is why there was a need for the widest possible range of knowledge regarding the monuments. Secondly, and this is especially true of Armenian heritage, which was deliberately destroyed in the territory of Turkey in the 1960s and 70s, the urgent documentary fixation of all preserved material was vital. Apparently, that is why the mission of the Italian scholars was presented as unquestionably noble, especially among the Armenian diaspora around the world, who enthusiastically welcomed each new exhibition.

The end product of collecting this information, as a result of the process of cataloging Armenian monuments, was the Milan series of books *Documenti di architettura armena*, showcasing magnificent design and high-quality color photographs. The accompanying texts were small and were often presented in the style of popular science. It is noteworthy that among these monographs is a work devoted to the Yereruyk Basilica—this is the second publication after Marr's book on this topic, detailing the many qualities of a unique monument of the 5th–6th centuries (Paboudjian et al. 1977). Later, in 2001, Italian specialists, together with colleagues from the Department for the Protection of Historical and Cultural Monuments of Armenia and with the support of the World Monument Fund, initiated a conservation program to preserve the majestic ruins of the basilica (Alpago Novello 2005, pp. 74–80).

The problems of restoring Armenian architectural monuments were largely the domain of the Milan group. More than half of the last exhibition, prepared by Alpago Novello in Venice, was devoted to the restoration of architectural monuments in Armenia and the participation of Italian specialists (Alpago Novello 2005).

The books published by the Roman group are also distinguished by their highly artistic style, but many of them are the result of long-term and fundamental scientific research—indeed, the word "research" was included in the title of the series. Some of the books were devoted to individual monuments, at the same time offering access to their typology and context. The rest of the series presented an analysis of some of the architectural types developed in the early Christian era. The idea of a "carpet" survey aimed at documenting this preserved heritage was abandoned in favor of a more focused and selective study (Bevilacqua and Gasbarri, p. 41).

The book published by Tommaso Breccia Fratadocchi about the church in Soradir not only introduced this monument, with its unusual plan, into the circle of research but also determined its place in the typology of tetraconchs with corner niches, forming a variation similar to that seen in the well-known Surb Khach church on Aghtamar Island that followed it (Breccia Fratadocchi 1971). Francesco Gandolfo was the first to investigate the whole set of single-nave hall churches, the most common type built in the early Christian era, showing the variety of their representations and clarifying their construction dates (Gandolfo 1973). Mario D'Onofrio monographically investigated a group of churches in the central quarter of Dvin, the capital of Armenia in the 5th–7th centuries. For the first time, the book included a combination of typological and iconographic analysis of the plans, along with a review of Dvin Cathedral, which acquired the form of a cross-domed triconch as a result of its rebuilding in the context of early Byzantine structures (D'Onofrio 1973). Previously, I tried to develop this line of research myself (Kazaryan 2012, vol. 2, pp. 425–34). As part of the series of books produced by the Roman group, Paolo Cuneo published a study of three particularly unusual hall churches located on the southern side of Lake Van, demonstrating the possibilities of the regional development of a particular architectural idea (Cuneo 1973).

These analytical studies include Paolo Cuneo's monograph on the "Ani Regional School of the Architecture of Armenia" (based on his report), which for the first time takes into account all the information gathered on the churches in the province of Shirak (see Figure 8), along with typological tables (Cuneo 1977). Unlike other regional schools, which have a clearly provincial character embodying their creative ideas, the Ani school of architecture has all the qualities that allow it to be considered metropolitan (Kazaryan 2017).

Various studies by my Italian colleagues have been published in scientific journals and conference collections. I would like to highlight a selection of articles that are valuable for the development of fundamental research, published in a volume of "Corsi di Cultura sull'Arte Ravennate e Bizantina 20" and dedicated to Armenian architecture. Among them is an article about the origin of the Armenian dome, written by Fernanda de' Maffei, an article on the church in Mastara, written by Breccia Fratadocchi, and another about the genesis of the domed hall, written by Armen Zarian.

Despite the fact that not a single major work has been written based on the results of all these studies, their cumulative impact on the course of development of Armenian architecture has been proven to be significant. They not only contributed competently processed information but also, to this day, influence the making of necessary adjustments to the methodology and theoretical views on the development of medieval architecture.

As a feature of de Frankovich's scientific method, my colleagues note its obvious formalism and, as a result, the expression of a certain distrust of the iconographic method, which was already widely used by prolific schools in the USA (Bevilacqua and Gasbarri 2020, p. 41). This same lag was characteristic of most Soviet architectural historians at that time. In Armenia and among the Moscow colleagues who studied Armenian architecture, a typological direction of research was developed, which adopted Strzygowski's method. However, by the end of the last century, a departure from their simplified views on the development of architecture was noticeable among young researchers analyzing Armenian

architecture, and the role of some Italian scholars is also significant in this regard. In the last quarter of the 20th century, architectural historians were faced with the task of identifying the most meaningful and emotional essence of a work of architecture and realizing the depth of creative ideas, which, in the near future, will perhaps be particularly necessary (Zekiyan 2016).

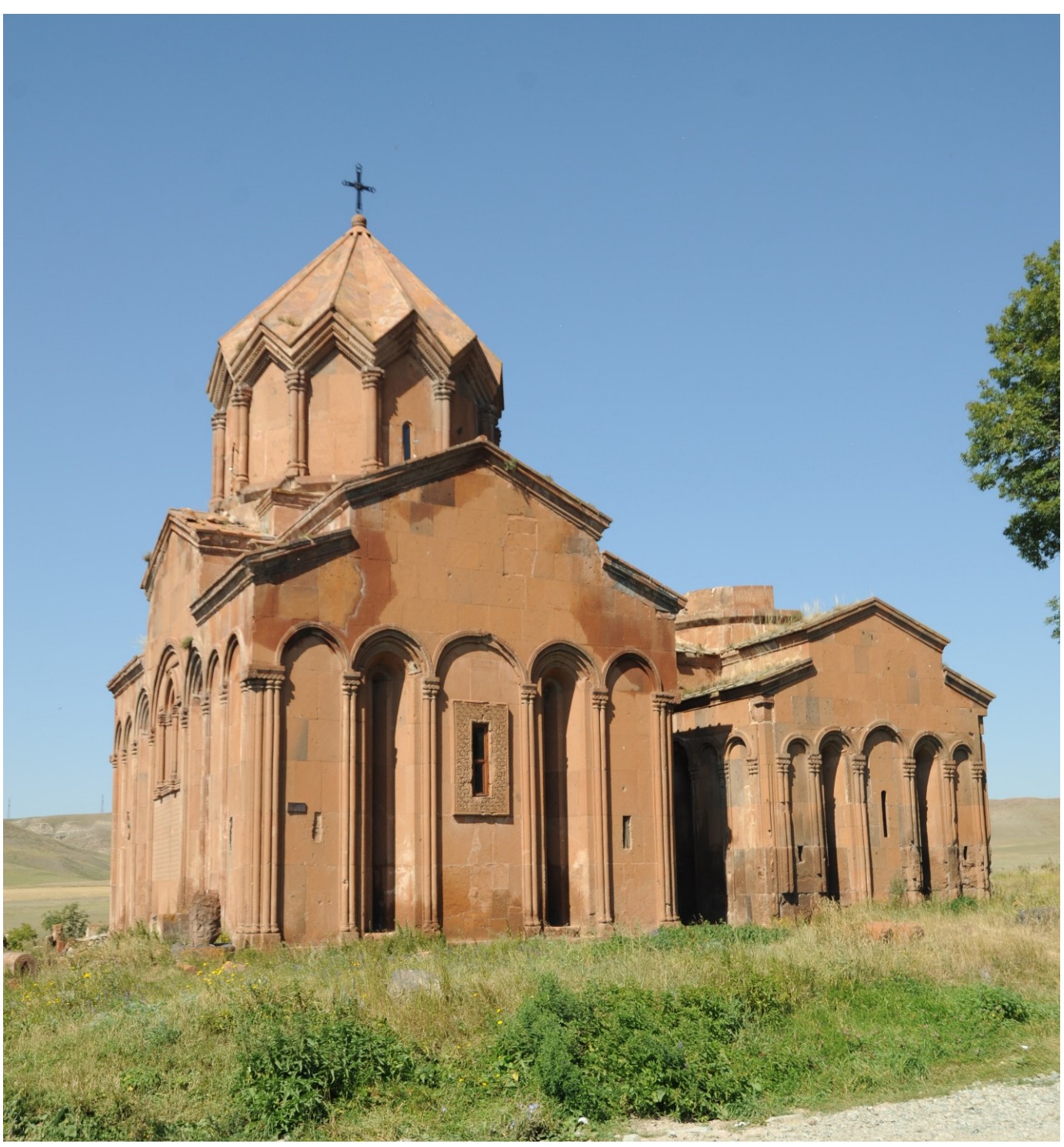

**Figure 8.** Marmashen monastery, showing the main group of buildings in the complex. First half of the 11th century. Photo by the author (2009).

**6. Results of This Joint Review of the Activities of the Two Missions and Concluding Observations**

A comparison of the nature and the results of the work of these two missions to explore the territory of historical Armenia leads to the identification of similarities and differences between the two missions, related to the opportunities they were offered, the historical and political realities, and the particular scientific interests of the group leaders. Among the missions' similar features, the following can be noted:

- Multi-year financing of the two projects, both from the state and from donations made by private individuals.
- The approximate number of participants and their professional levels.
- The collection of materials as a result of expedition trips, followed by their subsequent processing.
- The breadth of tasks and the variety of forms taken by the work associated with the projects.
- The combination of fundamental science objectives with practical tasks.
- The expansion of knowledge about these architectural monuments, and the creation of a collection of photographs.
- Qualitative changes in research directions and the projects' significant impact on the subsequent development of the history of Armenian architecture. Preservation of the research of Armenian architecture in the broad context of the development of medieval art was the most important contribution made by Russian and Italian scientists in their work on the monuments of Armenia and, in particular, Ani.
- The organization of serial publications based on the research results.
- The conducting of popular scientific work and its extensive coverage for the public, including the Armenian diaspora, along with information about the activities of these projects and new information about Armenian architecture.

Among the differences between the two missions are:

- The leading role played by orientalists from different fields of humanities in Marr's expedition and the leading role of architects in both groups of Italian scientists.
- The archeological objectives of the Ani Institute and their primary task of studying the architectural heritage explored in the expedition mounted by the Italians.
- The location of the Russian expedition in the medieval center of Armenia, in contrast to the absence of a permanent office for the Italian group anywhere in the historical Armenian territories.

The following two differences between the first mission and the second are related to this circumstance:

- The conducting of large-scale archeological excavations, as well as the group's strengthening buildings on their own initiative in the first case, and their participation in restoration activities commissioned by the state heritage protection bodies of Armenia in the second case.
- The creation of a museum by Marr, creating a permanent repository and exhibition of the work performed in Ani and the organization by the Italian groups of touring exhibitions of photographs showing the monuments, taken by Marr's Italian colleagues.
- The publication of the Institute's reports, serial brochures, and postcards produced by the Russian exhibition leaders, and the publication of monographs, album-type books, and postcards produced by the Italian groups.
- The organization of international symposiums by these Italian colleagues.

In fact, a century earlier Russian and later Italian not only archeological expeditions but also cultural missions entered into a form of time-stretched communication with each other on the basis of their mutual interest in Armenian art. Were there common grounds for such an interest to be found in the scientific communities of Russia and Italy, or, more profoundly, in the cultural traditions of the two peoples? This issue requires a separate

discussion. One can only be certain that their fascination with the countries of the East occurred in tandem with the orientalist currents moving through European culture, which have long been rooted in such scientific topics.

As was seen in the second half of the 19th century, the intensification of Byzantine studies in Russia at the turn of the 20th–21st centuries, and the need to restore their own traditions of studying the Christian East, contributed to the revival of comprehensive studies of Armenian art. Consistent state support for scientific initiatives in this research direction, including expeditions to the territories of Armenia and Turkey, have ensured good results. Ten years ago, research into Ani began again at NIITIAG, and now a group of ten scientists from the Moscow State University of Civil Engineering (MGSU), working under the guidance of the author of this article, is studying the architecture and monumental art of Ani. For the first time, the intensification of research by Russian and Italian scientists is taking place simultaneously, which allows us to raise the question of the possibility of future cooperation between groups of scientists from our two countries, who have been engaged in a common cause for decades. Is it possible to be involved in closer cooperation with groups of scholars from other countries such as Armenia, Turkey, and France, and to achieve the success seen during these earlier expeditions in our archeological and monument conservation works in Ani?[7] It is worth considering the forms that such cooperation might take since its prospects are obvious? I would like to conclude this article with a quotation from an essay on architecture written by one of the most active organizers of symposiums on Armenian art, now the Archbishop of Armenian Catholics of Istanbul and Turkey, L.B. Zekyan, in memory of a departed friend who was directly involved in the formation of Italian–Armenian collaborations and many other projects to support the foundation of cultural diplomacy. Zekyan wrote: "Gianclaudio Macchiarella was not only a connoisseur of art and architecture; he was also a diplomat who throughout his career as a cultural attache, from Tehran to Ankara, from Athens to Athena, always sought to intertwine relations between countries, nations and nationalities, paying special attention to the most complex relations, being convinced of the power of culture, its huge and, at times, untapped potential for dialogue" (Zekiyan 2016, p. 378)[8].

**Funding:** The study has been realized by a grant from the Russian Science Foundation, project no. 22-18-00354, https://rscf.ru/project/22-18-00354/ (accessed on 29 August 2023) in the Moscow State University of Civil Engineering (MGSU), National Research University.

**Data Availability Statement:** Not applicable.

**Conflicts of Interest:** The author declares no conflict of interest.

## Notes

[1] According to Leo, Kestner's work had a positive impact on the Catholicos Nerses Ashtaraketsi, who was on friendly terms with Vorontsov (Leo 1963, p. 40). On behalf of the Catholicos, in 1850, the epigraphy of Ani was copied by Vardapet Abel Mkhitarian, who published his work "Journey to Ani" in Constantinople.

[2] I thank the Academician Igor A. Bondarenko for relating his memories of meetings with P. Cuneo.

[3] This refers to the theoretical thoughts of Strzygowski (see: Bock 1983; Maranci 2001, pp. 79–175; Maranci 2001–2002).

[4] The photographs used in the exhibition were taken from the Scientific Archive of the Institute of the History of Material Culture, Russian Academy of Sciences: https://www.archeo.ru/struktura-1/nauchnyi-arhiv/vystavki/poetry-of-stones-ani-an-architectural-treasure-on-cultural-crossroads (accessed on 7 November 2023). About the exhibition at Depo in Istanbul see: https://www.anadolukultur.org/EN/34-our-works/179-poetry-of-stones-ani-an-architectural-treasure-on-cultural-crossroads-exhibition/ (accessed on 7 November 2023); the exhibition at Erimtan Archaeology And Arts Museum, see: https://erimtanmuseum.org/en/poetry-of-stones-ani-an-architectural-treasure-in-the-intersection-of-cultures (accessed on 7 November 2023), as well as in brochure: (Paludan-Müller et al. 2019).

[5] This refers to the impact of Italian scholars on the study of Armenian architecture (see: Maranci 2001, pp. 208–19).

[6] This references Marr's restoration of the Church of the Redeemer (Surb Prkich) in Ani (see: Kazaryan et al. 2016, para. 27–31).

[7] It is enough to mention the French expedition that worked in the 2000s; see, for example, Dangle (2017).

[8] Original text: "E Gianclaudio Macchiarella non fu solo uno studioso dell'arte e dell'architettura; fu anche un diplomatico che durante l'intera sua carriera come addetto culturale, da Teheran ad Ankara, da Atene a Washington, ebbe sempre di mira

l'intessere dei rapporti tra paesi, nazioni, popoli, rivolgendo una particolare attenzione alle relazioni più difficili, convinto com'era della forza della cultura, delle sue vaste e, talora, inopinate potenzialità di dialogo".

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
