# Peer review of "Architecture of Medieval Armenia as a Field of Research for Russian and Italian Scholars: Comparative Analyses of the Historiography"

_arts, 2023_

Round 1

Reviewer 1 Report

Comments and Suggestions for Authors

A large historiographical study of Armenian architecture and, in particular, the capital of medieval Armenia, Ani, has been undertaken. The original form of this research consists in a comparative analysis of two main schools of study of this heritage – the Russian, with its highest manifestation during the expedition of Nikolai Marr, and the Italian, with its several centers and main figures. Comparison of the results of archaeological and historical-architectural expeditions, exhibitions and publications reveals their similarities and differences, as well as their role in the development of knowledge on medieval architecture. In general, this attempt looks very successful, because earlier the roles of these scientific schools in the history of Armenian art were considered separately and without the connection between them. The article has a thoughtful and interesting structure. The conclusions are quite detailed. I recommend publishing this article if its author(s) takes into account my recommendations or part of them.

Recommendations and suggestions:

- illustrations are absent. The author can propose photos of the monuments of architectural, about which is a discussion in the article. As a result, the reader will be able to understand what kind of heritage is being discussed in this study;

- the article describes the work of Toros Toramanian and Josef Strzygowski in Ani. The bibliography contains researches on the activities of these researchers, but it would be correct to indicate their main own publications;

- in the title of the last section, enter please the word "conclusion". For example: "Conclusion: Results of the joint review of the activities of the two missions and concluding observations."

Author Response

1/ It recommended to insert some figures, photos of the monuments, which were been analyzed by the scholars.

2/ From review: “- the article describes the work of Toros Toramanian and Josef Strzygowski in Ani. The bibliography contains researches on the activities of these researchers, but it would be correct to indicate their main own publications”. Yes, thanks. I have to indicate their main publications. And I added the links to their books in the text, p.5. Also, the link to the book by Marr on the p. 6.

3/ I have added the word Conclusion in the last chapter’s title.

Reviewer 2 Report

Comments and Suggestions for Authors

- The article deals with the study and research campaigns on Armenian medieval architecture carried out by Russian and Italian expeditions, which have made it possible to enhance the value of Armenian medieval architectural heritage through the study, knowledge, research and dissemination of the artistic object. 

- This is a very elaborate, well-structured and interesting text that allows us to get to know the interest that existed, and still exists, in this subject, which is fundamental for the increasingly necessary understanding of Byzantine and Islamic art.

- The author has a rich and varied bibliography on these expeditions, the main subject of this work. For this reason, it would be advisable to reformulate the title so as not to give rise to any confusion, in which the main theme of the article would be more explicitly stated.

Author Response

1/ According the second review, “the main theme of the article would be more explicitly stated”.

If it relates with the necessity for the title’s edition, it can be enlarged as:

Architecture of the Medieval Armenia as a Field of Research of Russian and Italian scholars: Comparative Analyses of the Historiography

But this question must be solved by the editors of the thematic issue, especial by Anna Vyazemtseva.

2/ Also I have changed the title on the p. 5:

From the studies of the development of scholars’ expeditions

Brief historiographical overview

3/ Please, preserve this information into the article (I have insert it just after the conclusion and before the References):